# Gratitude to God Predicts Religious Well-Being over Time

**Philip Watkins** [1,*] , **Michael Frederick** [1] **and Don E. Davis** [2]

1   School of Psychology, Eastern Washington University, Cheney, WA 99004, USA; mfrederick4@eagles.ewu.edu
2   Department of Counseling and Psychological Services, Georgia State University,
    Atlanta, GA 30302-3980, USA; ddavis88@gsu.edu
*   Correspondence: pwatkins@ewu.edu

**Abstract:** The authors used a prospective design to investigate how gratitude to God predicts religious well-being over time. Gratitude to God is a central aspect of monotheistic religions, and thus may be particularly important to the religious/spiritual well-being of believers. Participants completed online measures of trait and state gratitude to God, along with spiritual well-being, nearness to God, and religious commitment scales over a one-to-two-month period. General well-being, trait gratitude, and the Big Five personality traits were also assessed. After controlling baseline levels, trait gratitude and the Big Five personality traits, dispositional gratitude to God at Time 1 predicted increased religious well-being, nearness to God, and religious commitment at Time 2. Although gratitude to God was significantly related to general well-being variables in cross-sectional analyses, it did not predict these variables over time. Validity data for the gratitude to God measures are also presented. The results suggest that gratitude to God is important to religious/spiritual well-being, and gratitude to God may be a critical variable for research on positive psychology and the psychology of religion/spirituality.

**Keywords:** gratitude; religiosity; spirituality; well-being

## 1. Introduction

> "We thank people for birthday presents of cigars and slippers. Can I thank no one for the birthday present of birth?"
>
> —G. K. Chesterton, *Orthodoxy* (Chesterton [1908] 1986, p. 258)

In the epigram above, Chesterton suggests one way that situations for experiencing gratitude might be increased: belief in a benevolent divine being who provides for a benefactor all the good things in life, and thus, there are many more blessings for which one can be thankful. But to date, few studies have investigated gratitude to God. Indeed, is divine gratitude a central aspect of spirituality and religiosity? The purpose of this study was to investigate the prospective relationship of gratitude to God to spirituality and religious well-being.

Why should gratitude to God be particularly beneficial to religious/spiritual well-being? In the following paragraphs, we describe theory foundational to this study. However, before presenting theory, we explain our critical concepts. We define the momentary state of gratitude as one's cognitive and emotional response to a circumstance one affirms as good, and which one largely attributes the benefit to an outside source. Thus, gratitude to God would be a response of gratitude to a benefit that one attributes at least, in part, to God. Trait, or dispositional gratitude, is one's tendency or disposition for gratitude. Thus, one high in trait gratitude would experience the state of gratitude frequently over a wide variety of circumstances. Therefore, trait gratitude to God would be one's disposition for experiencing gratitude to God. Those high in trait gratitude to God would have a low threshold for experiencing gratitude to God. Following Paloutzian and Ellison (1982), we define spiritual well-being as one's satisfaction with the spiritual domain of life, specifically,

satisfaction with one's relationship with the sacred or divine. In this study, we used three measures to assess religious/spiritual well-being: the religious well-being subscale of the Spiritual Well-Being Scale (Paloutzian and Ellison 1982); a revised version of the Nearness to God Scale (NTG-R; Gorsuch and Smith 1983; Uhder et al. 2010); and the Santa Clara Strength of Religious Faith Questionnaire (SRF; Plante and Boccaccini 1997).

In describing our theoretical approach, we begin with proposals about the factors that should contribute to religious/spiritual well-being. For monotheists at least, one's perception of one's relationship to God should be central to religious well-being. Furthermore, we submit that one's relationships with fellow believers should be important to religious well-being. If one's relationships with others in one's community of faith are troublesome, it stands to reason that that one will not be satisfied with the spiritual domain of one's life. We propose that when one lives in accord with one's spiritual/religious principles, one should be more satisfied with the spiritual dimension of one's life. One who is minimally committed to one's religious values would seem to live in discord, and thus should score lower in religious well-being.

But more specifically, how might gratitude to God contribute to religious well-being? In explaining our theoretical approach, we rely on several theories of general gratitude and apply them to gratitude to God. In their influential review, McCullough et al. (2001) proposed that gratitude is a moral affect in that it is a moral barometer (gratitude is a measure of how good others have been to us), a moral reinforcer (when one expresses gratitude, it reinforces moral behavior), and a moral motivator (when one feels grateful, one is motivated to act in a prosocial manner). How might this theory apply to gratitude to God? Firstly, gratitude to God might act as a moral barometer for one's relationship with the divine. Thus, when one feels grateful to God, one should feel favored and affirmed by God and feel that God is being good to them. When one experiences the goodness of God, it seems to follow that this would enhance one's perception of one's relationship with God, which, in turn, should enhance religious well-being. Secondly, if gratitude is a moral reinforcer in human relationships, then when one expresses gratitude to God, one should feel that God will provide more blessings for them in the future, at least at the subjective level. This increased hope regarding future blessings should also enhance one's perception of one's relationship with God, thus increasing one's religious well-being. Public expression of gratitude to God in one's community of faith should also bind relationships with other believers, thus enhancing religious well-being. Thirdly, if gratitude is a moral motivator in that it encourages prosocial behavior, gratitude to God should encourage moral behavior in the spiritual realm. Indeed, some have proposed that gratitude to God is a central motivator for religious behavior (e.g., Barth [1956] 1961; Keller 2015). Thus, gratitude to God should increase commitment to one's religious values, and thus support religious well-being.

Gratitude has been shown to be valuable to relationships. Algoe (2012) has theorized—with considerable empirical support—that gratitude helps one *find* new relationships, *reminds* one of important relationships, and helps *bind* healthy relationships. At least in terms of the *remind* and *bind* aspect of her theory, this would seem to easily translate to gratitude to God. When one experiences gratitude to God, this should remind one of the importance of one's relationship with the divine, which might promote relationship-enhancing behaviors (e.g., prayer, worship, and other devotional practices). Furthermore, gratitude to God should bind one in one's relationship with God in that one should experience a subjective appreciation of the goodness of God, which, in turn, should increase one's perceived intimacy with God. Taken together, it seems reasonable to propose that gratitude to God should enhance one's relationship with God.

In a complimentary theory, Watkins (2014) has argued that gratitude enhances subjective well-being because it amplifies the good in one's life. Just as an amplifier increases the signal strength of sound going into a microphone, gratitude amplifies one's awareness, memory, and experience of blessings in one's life. Translating this theory to the theme of this paper, gratitude to God should amplify one's awareness, memory, and experience of

the goodness of God to them. Consequently, this should increase one's perception of one's relationship with the divine, which should enhance religious well-being. In short, there is a good theoretical basis to propose that gratitude to God is important to religious and spiritual well-being.

Research has shown that gratitude enhances well-being (Wood et al. 2010; Watkins 2014). Gratitude is strongly associated with subjective well-being (McCullough et al. 2002; Watkins et al. 2003) and prospectively predicts well-being (e.g., Wood et al. 2008). Furthermore, numerous experimental studies have shown that gratitude exercises increase happiness (for reviews, see Davis et al. 2016; Watkins and McCurrach 2021). As described above, studies have also shown that gratitude enhances relational well-being (Algoe 2012). Thus, gratitude appears to be a critical facet of a good life. Although gratitude research has flourished in the last twenty years, few studies have investigated how the nature of the benefactor impacts gratitude.

"To whom are you grateful?" is a question that seems to have largely eluded gratitude researchers. Does it matter to whom one is grateful? If one feels grateful to a mafia boss, does this foster the same benefits as feeling grateful to a virtuous person such as Martin Luther King Jr.? Does the character and trustworthiness of the benefactor affect the experience of gratitude? These questions are particularly relevant when considering gratitude to God. Indeed, recent surveys show that the vast majority of Americans (87%) still believe in a benevolent God (Gallup 2019), and thus, divine gratitude may be a significant experience for many people.

It seems reasonable that God would be an important benefactor for religious people. People who believe in the God described by the Abrahamic religions—over 65% of religious people (Pew 2012)—believe that God is the ultimate source of all good. In a recent study, 100% of students who said that they believed in a personal God also said that God was an important benefactor in their life (Scheibe et al. 2017). Although many benefits come through human benefactors, most theists believe that these are "secondary causes" and that ultimately God is the source of blessing.

Furthermore, gratitude to God may be beneficial to theists because of what McCullough and colleagues referred to as *gratitude span* (McCullough et al. 2002). Gratitude span refers to the extent of the blessings that one might be grateful for and has been cited as one of the critical components of dispositional gratitude. Stated simply, grateful people have many things to be grateful for. One advantage of gratitude to God is that one can be grateful for any blessing, regardless of whether there is a perceived human benefactor, as illustrated by the Chesterton epigram. Thus, gratitude to God may enhance gratitude span. We propose that God is a significant benefactor for religious/spiritual people, and thus, gratitude to God may be vital to religiousness/spirituality. In summary, gratitude to God may be critical to the spiritual well-being of believers in God.

Research on gratitude to God has been scant, but the available evidence supports the idea that it deserves further investigation. Firstly, gratitude to God has been found to be associated with emotional and physical well-being (e.g., Krause et al. 2014; Krause et al. 2017). Secondly, it is well known that religious people tend to be happier than their unbelieving counterparts, but several recent studies have shown that gratitude to God mediates the relationship between religiosity and well-being (e.g., Rosmarin et al. 2011). Although these are promising results, these studies used cross-sectional designs which are weak and often misleading tests of mediation (Cole and Maxwell 2003). In the current study, we used a prospective design to better evaluate whether gratitude to God enhances religious well-being.

In summary, although gratitude enhances well-being, little is known about how the nature of the benefactor impacts gratitude. Most people believe in a benevolent God (Gallup 2019), and thus, God may be an important benefactor to many people, particularly those who are religious. Thus, gratitude to God is likely to be significant to the religious/spiritual well-being of many people. The purpose of this study was to investigate how gratitude to God prospectively predicts spiritual and religious variables over time. As the develop-

ment of valid measures of gratitude to God are needed for this area to progress, we also present psychometric data on a new state measure of gratitude to God, and cross-validate a previously developed measure of dispositional gratitude to God (Watkins et al. 2019). In the current study, participants completed our measures twice, one-to-two months apart. We administered our questionnaires assessing gratitude to God, along with measures of religiousness, spirituality, and general well-being scales. Our primary hypothesis was that that trait gratitude to God would predict increased religious well-being over time. As previous studies have shown small relationships between gratitude to God and general well-being scales, we conducted exploratory analyses using these variables.

## 2. Methods

### 2.1. Participants and Procedure

In this prospective design, 101 participants completed the measures one-to-two months apart (M days = 47.35). The questionnaires were administered online via SurveyMonkey, and thus, the time between administrations varied. Participants received partial course credit for completing each survey. Due to the anonymous nature of the study, we did not assess demographics. Four participants were eliminated from the analyses because they failed the data validity check item in either of our gratitude to God measures. These items were included to ensure the attention of participants and prevent random responding (e.g., "For the purpose of data checking, please endorse '2' for this item."). Three other measures were included for purposes not related to this study (two joy scales and a measure of social networks) and are not described below. After providing informed consent, participants completed the questionnaires in the order described below. This study was reviewed and approved by the Institutional Review Board of Eastern Washington University and adhered to the ethical principles of the American Psychological Association.

### 2.2. Measures

#### Positive and Negative Affect Scales (PANAS)

We used a modified version of the short PANAS (Watson et al. 1988). In addition to adjectives from the short PANAS, we also included adjectives from the Gratitude Adjectives Scale (GAS) (McCullough et al. 2002) ($\alpha$ = 0.88, 0.93); the Joy Adjectives Scale (Watkins et al. 2018); and descriptors we thought would be related to pride (e.g., "poised", "dignified") and narcissism ("superior", "better than others").

#### State Gratitude to God (S-GTG)

We developed a momentary state measure of gratitude to God that used comparative language (e.g., "Compared to how you usually feel, how grateful to God do you feel right now?"). This scale included 11 items, and participants responded to the statements on a 7-point scale ranging from "much less than I usually feel" to "much greater than I usually feel". As prior measures of state gratitude to God have shown the tendency to reveal both ceiling and floor effects (based on whether the participant was religious/spiritual), we hoped that this measure would eliminate these problems. Included in this measure were two reverse scored items, but unfortunately, these items had poor item-total correlations, and thus were deleted from the analysis. The final S-GTG showed excellent internal consistency ($\alpha$ = 0.97, 0.98) and as seen above, good construct validity (see Appendix A for this measure).

#### Santa Clara Strength of Religious Faith Questionnaire (SRF)

To measure religious commitment, we utilized the SRF (Plante and Boccaccini 1997). This is a relatively brief scale (9 items) that has been shown to effectively measure religious commitment ($\alpha$ = 0.98, 0.98).

*Satisfaction with Life Scale (SWLS)*

The SWLS is perhaps the most frequently used measure of the cognitive component of subjective well-being (Diener et al. 1985). A host of studies has supported the validity of this scale, and the SWLS showed good internal consistency in our study as well ($\alpha$ = 0.84, 0.91).

*Nearness to God Scale-Revised (NTG-R)*

The NTG-R was used to measure the perception of an individual's relationship with God (Gorsuch and Smith 1983). The original scale contained six items to which respondents were given a dichotomous forced choice ("agree" or "disagree"). In an attempt to improve the psychometrics of the scale, we added four items and changed the response scale to a 7-point scale ranging from "completely disagree" to "completely agree". This scale has shown good reliability and validity (Uhder et al. 2010) and had excellent internal consistency in this study ($\alpha$ = 0.96, 0.94).

*Spiritual Well-Being Scale*

We used the Spiritual Well-Being Scale (Paloutzian and Ellison 1982). This measure contains two subscales: existential well-being ($\alpha$ = 0.86, 0.88) and religious well-being ($\alpha$ = 0.95, 0.96). Participants respond to items such as "I believe that God loves me and cares about me" on a five-point scale ranging from "Strongly Agree" to "Strongly Disagree".

*Big Five Inventory-2 Extra Short Form (BFIxs)*

The use of the BFIxs gave us the ability to control the Big Five personality characteristics in our hierarchical regression analyses. The BFIxs (Soto and John 2017) contains 15 items that provide a good assessment of the Big Five personality characteristics. In prospective studies such as this, it is important to control basic personality traits, and this measure served that function.

*Flourishing Scale (FS)*

The FS was developed as a brief measure of eudaimonic well-being (Diener et al. 2010) and served as one of our measures of general well-being. There are eight items in the FS, and participants respond to statements such as "I lead a purposeful and meaningful life" on a seven-point agree/disagree scale ($\alpha$ = 0.90, 0.93).

*Gratitude Questionnaire (GQ-6).*

The GQ-6 served as one of our measures of dispositional gratitude (McCullough et al. 2002). This is perhaps the most utilized measure of trait gratitude and has good psychometric properties supporting its use ($\alpha$ = 0.86, 0.84).

*Gratitude, Resentment, and Appreciation Test-Short (GRAT-S).*

The GRAT-S is a short version (16 items) of the GRAT (Watkins et al. 2003) and has been used frequently to assess trait gratitude. This measure approximates the full GRAT well and contains good psychometric properties (Thomas and Watkins 2003) ($\alpha$ = 0.89, 0.91).

*Subjective Happiness Scale (SHS)*

The SHS was developed to provide a more affectively loaded measure of happiness than the SWLS (Lyubomirsky and Lepper 1999). This measure is frequently used in positive psychology research and served as our third measure of general well-being ($\alpha$ = 0.87, 0.87).

*Trait Gratitude to God Scale (GTG-T).*

The GTG-T was developed to measure one's disposition for experiencing gratitude to God, and thus, it serves as the primary predictor variable in this study. This scale contains 10 items (e.g., "God has given me an overwhelming number of blessings in my life") that participants respond to on a 9-point scale ranging from "strongly disagree" to "Strongly agree". Previous research has shown that this measure has excellent psychometric

properties (Uhder et al. 2010; Watkins et al. 2019); however, in the current study the reverse scored item of the GTG-T did not correlate well with the total score. Thus, we eliminated this item when computing our scale means, and this somewhat revised scale showed excellent internal consistency ($\alpha = 0.99, 0.99$). After participants completed this scale, they were thanked for their participation and awarded their course credit.

## 3. Results

### 3.1. Preliminary Analyses

Correlations among variables are reported in Table 1.[1] Trait gratitude to God, as measured by the Trait Gratitude to God Scale (GTG-T), was very strongly related to religious well-being, strength of religious faith, and nearness to God. Test-retest correlations showed that both the Trait- (GTG-T) and the State Gratitude to God (S-GTG) Scales were stable across time (GTG-T r = 90; S-GTG r = 0.71). As expected, the trait measure of gratitude to God was more stable than the state scale. In Table 1, we also show associations with the Big Five personality traits as assessed by the Big Five Inventory-2 Extra-Short form (BFIxs). These correlations show that people who tend to be grateful to God tend to be somewhat extraverted, agreeable, and emotionally stable. See Table 2 for means and SDs at Time 1.

### 3.2. Hierarchical Regression Analyses

To investigate how Time 1 trait gratitude to God predicted changes in our outcome variables, we used a two-step hierarchical regression predicting the Time 2 outcome variable. In Step 1, we entered scores on the Time 1 outcome variable, along with Time 1 scores of the Big Five personality traits and trait gratitude (GQ-6 and GRAT-S). In Step 2, we entered Time 1 scores of trait gratitude to God as measured by the Gratitude to GTG-T. This analysis allowed us to investigate how Time 1 trait gratitude to God predicted the Time 2 outcome variables after controlling the Time 1 variables.

Firstly, we present analyses of our predicted outcome variables, followed by the exploratory analyses. Trait gratitude to God should predict increases in state gratitude to God over time. Model 1 was significant ($F(8,92) = 13.80$, $p < 0.001$), and as predicted, the $F_{change}$ for Model 2 was also significant ($F(1,91) = 20.17$, $p = 0.01$, $\Delta R^2 = 0.082$, GTG-T standardized $\beta = 0.389$, partial correlation = 0.426). Thus, GTG-T scores at Time 1 predicted increased S-GTG scores at Time 2, providing some evidence of the validity of both measures.

Time 1 trait gratitude to God predicted increases in the strength of religious commitment as measured by the SRF. Again, Model 1 was significant ($F(8,93) = 76.47$, $p < 0.001$), but importantly, the $F_{change}$ for Model 2 was also significant ($F(1,92) = 10.90$, $p = 0.001$, $\Delta R^2 = 0.014$, GTG-T standardized $\beta = 0.236$, partial correlation = 0.325). Hierarchical regression also showed that the GTG-T predicted increases in one's felt nearness to God. Model 1 for this analysis was significant ($F(8,93) = 32.42$, $p < 0.001$), and the $F_{change}$ for Model 2 was also significant ($F(1,92) = 21.79$, $p < 0.001$, $\Delta R^2 = 0.051$, GTG-T standardized $\beta = 0.418$, partial correlation = 0.438). A similar effect was found regarding the religious well-being scale from the Spiritual Well-Being Scale. Again, Model 1 for this analysis was significant ($F(8,92) = 53.14$, $p < 0.001$), and the $F_{change}$ for Model 2 was significant ($F(1,91) = 24.67$, $p < 0.001$, $\Delta R^2 = 0.038$, GTG-T standardized $\beta = 0.416$, partial correlation = 0.462). However, the hierarchical regression for the existential well-being scale was not significant. Although Model 1 for this analysis was significant ($F(8,92) = 17.11$, $p < 0.001$), the $F_{change}$ for Model 2 was not ($F(1,91) = 0.07$, $p = 0.795$, $\Delta R^2 < 0.001$, GTG-T standardized $\beta = -0.019$, partial correlation = $-0.027$). In summary, dispositional gratitude to God predicted increases in variables related to religiosity and spiritual well-being, except for existential well-being.

**Table 1.** Correlations of Time 1 Variables.

| Scale | 1 | 2 | 3 | 4 | 5 | 6 | 7 | 8 | 9 | 10 | 11 | 12 | 13 | 15 | 16 | 17 |
|---|---|---|---|---|---|---|---|---|---|---|---|---|---|---|---|---|
| 1. GTG-T | – | | | | | | | | | | | | | | | |
| 2. S-GTG | 0.70 ** | – | | | | | | | | | | | | | | |
| 3. GQ-6 | 0.36 ** | 0.40 ** | – | | | | | | | | | | | | | |
| 4. GRAT-S | 0.34 ** | 0.311 ** | 0.74 ** | – | | | | | | | | | | | | |
| **SWBS** | | | | | | | | | | | | | | | | |
| 5.-RWB | 0.85 ** | 0.68 ** | 0.29 ** | 0.25 ** | – | | | | | | | | | | | |
| 6.-EWB | 0.15 * | 0.19 * | 0.69 ** | 0.52 ** | 0.21 * | – | | | | | | | | | | |
| 7. SRF | 0.85 ** | 0.68 ** | 0.34 ** | 0.30 ** | 0.84 ** | 0.19 * | – | | | | | | | | | |
| 8. NTG-R | 0.84 ** | 0.71 ** | 0.33 ** | 0.27 ** | 0.87 ** | 0.19 * | 0.87 ** | – | | | | | | | | |
| 9. GAS | 0.35 ** | 0.43 ** | 0.46 ** | 0.35 ** | 0.30 * | 0.30 ** | 0.32 ** | 0.33 ** | – | | | | | | | |
| 10. SWLS | 0.14 | 0.19 * | 0.56 ** | 0.50 ** | 0.11 | 0.68 ** | 0.12 | 0.13 | 0.34 ** | – | | | | | | |
| 11. SHS | 0.30 ** | 0.24 * | 0.64 ** | 0.54 ** | 0.24 * | 0.66 ** | 0.28 ** | 0.28 ** | 0.39 ** | 0.60 ** | – | | | | | |
| 12. FS | 0.23 * | 0.28 * | 0.75 ** | 0.60 ** | 0.17 * | 0.74 ** | 0.23 ** | 0.23 * | 0.36 ** | 0.63 ** | 0.64 ** | – | | | | |
| **Big Five Traits** | | | | | | | | | | | | | | | | |
| 13. Ext | 0.17 * | 0.13 | 0.18 * | 0.09 | 0.20 * | 0.26 ** | 0.17 * | 0.20 * | 0.12 | 0.31 ** | 0.35 ** | 0.38 ** | – | | | |
| 14. Agreeable | 0.15 * | 0.18 * | 0.36 ** | 0.41 ** | 0.11 | 0.30 ** | 0.15 * | 0.09 | 0.14 * | 0.20 * | 0.37 ** | 0.34 ** | 0.04 | – | | |
| 15. Consc | 0.12 | −0.02 | 0.34 ** | 0.22 * | 0.05 | 0.38 ** | 0.02 | 0.02 | 0.21 * | 0.33 ** | 0.33 ** | 0.50 ** | 0.24 ** | 0.14 | – | |
| 16. Neurot | −0.20 * | −0.14 * | −0.44 ** | −0.35 ** | −0.13 | −0.51 ** | −0.22 * | −0.16 * | −0.23 ** | −0.48 ** | −0.63 ** | −0.52 ** | −0.31 ** | −0.20 * | −0.35 ** | – |
| 17. Openness | 0.05 | 0.16 * | 0.26 ** | 0.26 ** | −0.001 | 0.14 | 0.04 | 0.04 | 0.20 * | 0.23 ** | 0.07 | 0.21 * | 0.21 * | 0.12 | −0.02 | −0.03 |

Note: GTG-T = Trait Gratitude to God; S-GTG = State Gratitude to God; GQ-6 = Gratitude Questionnaire; SWBS = Spiritual Well-Being Scale; RWB = Religious Well-Being Subscale of the SWBS; EWB = Existential Well-Being Subscale of the SWBS; GAS = Gratitude Adjectives Scale (a state gratitude measure); SRF = Strength of Religious Faith Scale; NTG-R = Nearness to God Scale-Revised; SWLS = Satisfaction with Life Scale; SHS = Subjective Happiness Scale; FS = Flourishing Scale; Ext = Extraversion; Consc = Conscientiousness; Neurot = Neuroticism. * $p < 0.05$; ** $p \leq 0.001$.

**Table 2.** Means and Standard Deviations at Time 1.

| Variable | Mean | SD | N |
| --- | --- | --- | --- |
| GTG-T | 5.22 | 2.83 | 188 |
| S-GTG | 4.07 | 1.50 | 193 |
| GQ-6 | 5.73 | 1.03 | 190 |
| GRAT-S | 7.00 | 1.16 | 190 |
| RWB (SWBS) | 3.35 | 1.53 | 191 |
| EWB (SWBS) | 4.26 | 0.94 | 192 |
| SRF | 2.22 | 0.97 | 193 |
| NTG-R | 3.89 | 1.86 | 193 |
| GAS | 3.64 | 0.95 | 196 |
| SWLS | 4.35 | 1.29 | 193 |
| SHS | 4.55 | 1.34 | 191 |
| FS | 5.44 | 1.00 | 190 |
| **Big Five Traits** | | | |
| Extraversion | 3.07 | 0.89 | 193 |
| Agreeable | 3.77 | 0.71 | 193 |
| Conscientiousness | 3.40 | 0.82 | 193 |
| Neuroticism | 3.33 | 0.91 | 193 |
| Openness | 3.80 | 0.74 | 193 |

Note: Means for the scales are average/item. GTG-T = Trait Gratitude to God; S-GTG = State Gratitude to God; GQ-6 = Gratitude Questionnaire; SWBS = Spiritual Well-Being Scale; RWB = Religious Well-Being Subscale of the SWBS; EWB = Existential Well-Being Subscale of the SWBS; GAS = Gratitude Adjectives Scale (a state gratitude measure); SRF = Strength of Religious Faith Scale; NTG-R = Nearness to God Scale-Revised; SWLS = Satisfaction with Life Scale; SHS = Subjective Happiness Scale; FS = Flourishing Scale.

Due to the potential for collinearity of our trait gratitude measures at Step 1, we conducted the identical hierarchical regression analyses as above but excluded our trait gratitude measures from Model 1. These analyses for our spiritual well-being measures were similar to our planned analyses described above. All Model 2 $F_{change}$ scores were significant (all ps $\leq$ 0.002), and partial correlations ranged from 0.32 to 0.45.

Observation of the items from the existential well-being scale of the Spiritual Well-Being Scale shows that this is largely a measure of general well-being. As will be seen, the finding on the hierarchical regression for this measure was consistent with those of our other general well-being questionnaires. Indeed, hierarchical regression analyses for the Gratitude Adjectives Scale (GAS), Satisfaction with Life Scale (SWLS), Subjective Happiness Scale (SHS), and Flourishing Scale (FS) all showed that Model 2 was nonsignificant (all $F_{change}$ ps > 0.61, all $\Delta R^2$ < 0.002). Thus, trait GTG-T scores at Time 1 did not predict reliable increases in general well-being scales at Time 2.

## 4. Discussion

Recent work has shown that religious/spiritual well-being serves as a protective factor against the deleterious effects of the COVID-19 pandemic (González-Sanguino et al. 2020). Thus, research on factors that contribute to religious well-being is particularly timely. In this study, we found that dispositional gratitude to God predicted increased religiosity/spirituality over time, supporting the idea that gratitude to God is a significant factor for religious well-being. After controlling Time 1 baseline levels, trait gratitude, and the Big Five personality traits, we found that levels of trait gratitude to God at Time 1 positively predicted Time 2 strength of religious commitment, one's perception of nearness to God, and religious well-being. Although gratitude to God was related to general well-being variables in cross-sectional analyses, it did not predict changes in these variables over time. In the discussion below, we explore the theoretical and practical implications of these findings.

Our results suggest that gratitude to God is important to religious well-being, and these appear to be consistent with past findings (e.g., Rosmarin et al. 2011; Watkins et al. 2019). Why is divine gratitude important to religious believers? Research on general gratitude may be informative to this question. As mentioned earlier, studies have shown

that gratitude enhances human relationships (Algoe 2012; Bartlett et al. 2012). Thus, it seems reasonable to propose that gratitude to God should enhance one's relationship with God. Indeed, we found that dispositional gratitude to God predicted enhanced intimacy with God over time. It is also possible that gratitude to God decreases doubt about the existence of God, thus increasing religious commitment and a sense of nearness to God. Indeed, Watkins et al. (2019) found that gratitude to God predicted increased confidence in the existence of God over time, and more research could be devoted to investigating how divine gratitude impacts doubts about the existence of God.

Another explanation for how gratitude to God is important to religious well-being was initially suggested by McCullough and colleagues (McCullough et al. 2001; McCullough et al. 2002). They proposed that those who believe in God always have a benevolent benefactor to thank for benefits. Thus, for religious believers, all benefits can potentially become gifts—divine favors intentionally provided to the believer. As Chesterton opined, "All goods look better when they look like gifts" (Chesterton [1924] 1990, p. 78). Is a benefit experienced more positively when it is perceived as a gift? Some evidence suggests that it does. Watkins et al. (2013) showed that individuals enjoyed a benefit more when it was presented as a gift rather than as a good (an equivalent benefit not intended to help the beneficiary). In short, a benefit was experienced with more happiness when it looked like a gift rather than a mere good. As this was a scenario study, more research is needed to investigate the proposition of Chesterton, but it is somewhat surprising that studies have rarely investigated this idea because this might be a notable mechanism that helps us understand the gratitude and well-being relationship.

As all goods can become gifts for those who believe in a benevolent God, this should increase their gratitude span, which should enhance well-being. Gratitude span refers to the extent of blessings that one might be grateful for (McCullough et al. 2002). Belief in a benevolent God makes all blessings potentially things that one might be grateful for, thus enhancing the gratitude span. The more things that one is grateful for, the more grateful one should be, and research has definitively shown that the more grateful one is, the happier one tends to be (McCullough et al. 2002; Watkins et al. 2003). Taken together, another potential explanation for the benefit of gratitude to God to spiritual/religious well-being might be that gratitude to God enhances the span of gratitude.

Our theoretical analysis of how gratitude to God supports religious well-being implies that gratitude to God is of particular importance. However, believers in God likely experience many other positive emotions toward God, such as hope, joy, interest, and awe. Is gratitude to God of particular importance to religious/spiritual well-being, or is it simply one of many positive emotions that support a spiritual satisfaction of believers? Our data cannot provide an answer to this question, but future research could focus on this issue.

If divine gratitude is important to religious well-being, this has noteworthy practical implications. Increasing gratitude to God should result in increased religious well-being. How might gratitude to God be encouraged? What spiritual practices might enhance gratitude to God? What we know from the general gratitude literature could be applied to divine gratitude. For example, a regular practice of grateful recounting (also known as gratitude lists) might be modified for focusing on God as a benefactor. An individual might recall three blessings and reflect on how God was involved with providing these benefits. Other practices such as gratitude letters or grateful reappraisal of unpleasant events could be adapted to a divine benefactor. Time-honored spiritual practices such as prayer, contemplation of sacred texts, and personal or corporate worship might be even more effective in enhancing gratitude toward God. Of course, these practices might only be of interest to individuals of faith, but it seems that this would be a fruitful area for future research.

Our results also provide validity data for two gratitude to God measures. The Trait- (GTG-T) and State Gratitude to God Scales (S-GTG) showed good evidence of reliability and validity. Both measures showed excellent internal consistency (ranging from 0.97 to 0.99), and good evidence of construct validity. As expected, Time 1 trait gratitude to God

predicted increased Time 2 state gratitude to God, providing evidence of the validity of both measures. In addition, as predicted, both measures showed strong relationships to religious variables, and were more closely associated with these scales than trait gratitude (see Table 1). In fact, some of these relationships were so strong that concerns might be raised about whether these are distinct constructs. However, even with these high correlations, the GTG-T still predicted increases in critical variables after controlling levels of the Time-1 religious variable.

As religiosity has often been found to be related to general subjective well-being, we would expect that gratitude to God should be related to these variables, and indeed, this was the case in this study. Stronger associations were found between gratitude to God and general trait gratitude, and these correlations add evidence of the validity of our gratitude to God measures. In summary, this study provides preliminary evidence for the validity of our state and trait gratitude to God questionnaires (see also Watkins et al. 2019), and researchers in positive psychology and the psychology of religion may find these scales useful.

Several limitations of this study should be highlighted. Of course, the well-known problems with self-report apply to this study. One could argue that biases created by self-report are even more of an issue with religious variables because religious individuals would like to see themselves in a favorable spiritual light. Future studies should attempt to control self-presentation biases. That being said, in our critical analyses, we controlled Time 1 baseline levels of the outcome variables, thereby essentially controlling self-presentation bias. We believe that future studies should recruit a population that is more religiously diverse than that of this study. In other studies, we have found that students at this university show higher rates of atheism than is reflective of the population more generally. Thus, we tend to see individuals that are either religiously committed or show some antipathy toward religion. This might explain why we achieved such high correlations between our religious/spiritual variables, and future studies may want to recruit a more diverse religious sample.

In summary, dispositional gratitude to God predicted increased religious well-being, intimacy with God, and religious commitment over time. The results from this study suggest that gratitude to God is important to religious well-being. This may be because divine gratitude supports all benefits being seen as gifts. "The great painter boasted that he mixed all his colours with brains", observed Chesterton, "and the great saint may be said to mix all his thoughts with thanks" (Chesterton [1924] 1990, p. 78). Indeed, gratitude to God may provide religious/spiritual people with a way to mix all their thoughts with thanks, which consequently increases their religious well-being.

**Author Contributions:** Conceptualization, P.W. and D.E.D.; methodology, P.W., M.F. and D.E.D.; software, M.F.; validation, P.W., M.F. and D.E.D.; formal analysis, P.W.; writing—original draft preparation, P.W.; writing—review and editing, P.W., M.F. and D.E.D.; supervision, P.W.; project administration, P.W.; funding acquisition, P.W. and D.E.D. All authors have read and agreed to the published version of the manuscript.

**Funding:** This research was funded by the John Templeton Foundation, "The Form and Function of Gratitude to God", grant number G21000019.

**Institutional Review Board Statement:** The study was conducted in accordance with the Declaration of Helsinki, and approved by the Institutional Review Board of Eastern Washington University (protocol code HS-5901 and date of approval: 4 May 2020).

**Informed Consent Statement:** Informed consent was obtained from all subjects involved in the study.

**Data Availability Statement:** The data that support the findings of this study are openly available at https://osf.io/kptju/ (accessed on 16 June 2022).

**Conflicts of Interest:** The authors declare no conflict of interest.

**Appendix A**

**S-GTG**

For the following items, please respond as to *how you're feeling right now, in this moment.* Simply circle the response that best represents how you're feeling right now, compared to how you usually feel.

1. Compared to how you usually feel, how grateful to God do you feel right now?

| 0 | 1 | 2 | 3 | 4 | 5 | 6 |
|---|---|---|---|---|---|---|
| Least grateful to God than I've ever felt | | | About what I usually feel | | | Most grateful to God I've ever felt |

2. Compared to how you usually feel, how much do you feel that God has provided you with an abundant life? (i.e., plenty of everything)

| 0 | 1 | 2 | 3 | 4 | 5 | 6 |
|---|---|---|---|---|---|---|
| Least abundant life that I've felt | | | About what I usually feel | | | Most abundant life that I've ever felt |

3. Compared to how you usually feel, how much are you experiencing the goodness of God right now?

| 0 | 1 | 2 | 3 | 4 | 5 | 6 |
|---|---|---|---|---|---|---|
| I feel much less of the goodness of God | | | About what I usually feel | | | I feel the goodness of God more than ever |

4. Compared to how you usually feel, how much do you feel a warm sense of appreciation toward God right now?

| 0 | 1 | 2 | 3 | 4 | 5 | 6 |
|---|---|---|---|---|---|---|
| Much less than I usually feel | | | About what I usually feel | | | Much greater than I usually feel |

5. Compared to how you usually feel, how generously do you feel God has treated you?

| 0 | 1 | 2 | 3 | 4 | 5 | 6 |
|---|---|---|---|---|---|---|
| Much less than I usually feel | | | About what I usually feel | | | I feel God's generosity more than ever |

6. Compared to how you usually feel, how thankful to God do you feel right now?

| 0 | 1 | 2 | 3 | 4 | 5 | 6 |
|---|---|---|---|---|---|---|
| Much less thankful than usual | | | About what I usually feel | | | More thankful to God than I've ever felt |

7. Compared to how you usually feel, how much are you experiencing the grace of God?

| 0 | 1 | 2 | 3 | 4 | 5 | 6 |
|---|---|---|---|---|---|---|
| Much less than I usually feel | | | About what I usually feel | | | I feel God's grace more than ever |

8. Compared to how you usually feel, how much do you feel you have to thank God for?

| 0 | 1 | 2 | 3 | 4 | 5 | 6 |
|---|---|---|---|---|---|---|
| Much less than I usually feel | | | About what I usually feel | | | Much greater than I usually feel |

9. Compared to how you usually feel, how thankful are you for all the people God has brought into your life?

| 0 | 1 | 2 | 3 | 4 | 5 | 6 |
|---|---|---|---|---|---|---|
| Much less than I usually feel | | | About what I usually feel | | | More thankful to God for others than I have ever felt |

## Note

[1]  Correlations at Time 2 showed essentially the same pattern as the relationships at Time 1.

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
