# Peer review of "Gratitude to God Predicts Religious Well-Being over Time"

_religions, doi:10.3390/rel13080675_

Round 1

Reviewer 1 Report

I am not sure this article provides any illumination about the psychological processes that mediate spiritual well-being.  The abstract and the body of the text say that gratitude to God is important in SWB, i.e., scores on gratitude toward God predict scores on the religious well-being subscale of the SWBS.  Duh.  This is not exactly a revelatory finding.  It means only that a good feeling about and positive attitude towards God is correlated positively with a set of statements that reflect positive feelings and affirmative attitudes toward God, and is correlated with another set of statements that are stated in the inverse, negative direction about feelings about and attitudes towards God.  I.e., a positive reflection about God predicts positive reflections about God.  So I don't think it adds anything to our psychological understanding of the processes that mediate the obtained relationships to say that a positive feeling about God predicts positive feelings about God.    What would have been psychologically interesting is if the results had come out the other way. 

As to the other relationship key to what is said in the abstract and the body of the text (i.e., that gratitude to God doesn't predict much of what the authors call "secular well-being"; or as far as the SWB scale itself is concerned, this presumably refers in part to the Existential Well-Being subscale of the SWBS), one must ask, "Why, exactly, ought we suspect that gratitude to god would predict EWBS?". If the concept "God" is not relevant to someone, then it presumably oughtn't be associated with non-God items or content in any particular way.  To the contrary, theoretically the correlations between gratitude to God and the EWB or "secular" items ought to be lower and closer to neutral overall (just as the correlations between the RWBS and EWBS are typically lower and closer to neutral than they are to other variables). And that is how the data come out.  Again, this finding is not psychologically interesting. A real trigger to future in-depth research would have happened if the findings had come out the other way. 

I think it would need to be a major revision that delves deeply into the theoretical bases that underpin the findings, and propose alternative study approaches to uncover the processes that lead to discrepancies in findings, instead of the "A predicts A" kind of finding that this research presents. That would make it a really interesting psychology article.

Author Response

R1 was basically concerned about the significance of the findings. Yes, if the trait gratitude to God measure (GTG-T) simply “correlated” with measures of spiritual well-being, we would have to agree that the findings would be of limited importance. However, this was not a cross-sectional correlational design, but a prospective design with two waves of data. Prospective designs are much more powerful than cross-sectional designs because they rule out various concerns such as operational confounds (two measures inadvertently measuring the same construct), several third variable issues (such as person and environmental confounds), and time sequence issues. For example, our prospective design rules out R1’s concern that our findings simply reveal that one’s positive responses to one set of statements correlate with their responses to another set of statements, because our prospective design lets us control for various self-report biases and operational confounds at Time 1, thus allowing us to conclude that GTG-T actually predicts increases in our spiritual well-being variables over time. We agree that in some ways this is a preliminary finding, but it is much more than simply GTG correlating with people’s report on spiritual wellbeing measures. We agree with R1 that the paper needs more theory development, and we appreciate the opportunity to provide that development in our introduction, which we include just before paragraph 2 in the Introduction.

Reviewer 2 Report

Good contribution to an important topic, that is well situated in the discussion. Theory can be enhanced however: the empirical correlation of scales as such is not a theory, so the article could benefit from theoretical input (before paragraph 2) preceding empirical analysis. Thus for instance, the inclusion of personality scales is hardly argued whereas other likely variables - for instance those that could clarify current misfortune - are mssing.

The empirical part seems to be fine, however visual gaphs could be helpful to the reader. Cross-validity of scales is warranted. The found correlations (over time) of gratitude towards God and spiritual wellbeing is rather obvious as both variables - and other spiritual scales - belong to the same conceptual complex and do require theoretical demarcation when included in empirical research. SEM analyses could put the implicit causal interrelations of the various aspects of spirituality tot the test.

Author Response

We agree with R2 that more theory development is needed, and we have followed their recommendation for providing this before paragraph 2. We appreciate this recommendation and hope that this theory development improves the paper. We believe that our original hierarchical regression analyses speak to R2’s concern about drawing causal inferences, and we hope that the theoretical development provided adds some meat to our findings.

Reviewer 3 Report

I read the paper “Gratitude to God is important to Spiritual Well-Being” with great interest. This is a nice compact study that appropriately uses a repeated measures design to examine the prediction of gratitude to God on outcomes over time. My comments are made with the goal of helping the authors further strengthen the paper.

1.       Sample Descriptives: I encourage the authors to include information about the sample. What are the demographics (e.g., age, race, gender), religiosity levels, and religion? This is important in helping understand the generalizability of the results.

2.       Acronyms: It would be helpful to ensure that all the acronyms in the paper are elaborated on (e.g., SRF) in the text.

3.       GRAT-S scale: GRAT-S was used in the analysis but not included in the Table of correlations. I encourage the authors to include this in the correlations. Also, could the authors also provide the means and SDs in the correlation table of variables so readers can understand what the levels of these variables might be?

4.       Analysis: Regarding the analysis, it is unclear why two measures of trait gratitude were needed as controls. Multicollinearity can lead to problems with the parameter estimates in the regression model. Are the results robust when controlling for only one or another trait gratitude?

5.       Conclusions: While I believe that the conclusions drawn are appropriate, I wonder whether the authors can be a little more tempered in their conclusions (abstract and title) given the small sample size (~100 individuals) and questions about external validity, especially when these results are not replicated in a larger sample.

Wishing the authors the best in this work!

Author Response

  1. As this was an anonymous study, we did not assess demographics. This was probably an oversight on our part, but if needed, we could provide comparable demographics of other studies that we ran at the same time as this study in the same population, if this were to be helpful.
  2. We have gone through the manuscript again to clarify our acronyms, per APA style.
  3. We have now added the GRAT-S to our correlation table, as recommended by R3. Because the correlation table was already large, we elected to add another table describing the means and SDs, as asked for by R3 (Table 2).
  4. As recommended by R3, we conducted the hierarchical regression without the trait gratitude measures. The results were similar to the initial analyses, and we provide the description of this additional analysis in the Results.
  5. We have also reviewed the manuscript and tried to temper our claims (and our title) somewhat, as recommended by R3. Hopefully this improves the paper.

Round 2

Reviewer 1 Report

I think that for this paper to be published, the author needs to change an important use of terms throughout the paper.  The current text continues to say that gratitude to God predicts higher Spiritual Well-Being (SWB).  This is said in its generic terminology over and over. But it is misleading.  The data show that gratitude to God predicts higher Religious Well-Being (RWB), i.e., higher scores on the RWB subscale of the SWB Scale.  So when the author keeps on saying that higher gratitude to God predicts higher SWB, it is only a partially accurate statement and, frankly, is misleading.  It is misleading because total SWB is comprised of RWB and EWB. But the text is clear in saying that higher gratitude to God does not predict higher EWB longitudinally.  So it is not total SWB, but scores on the RWB subscale, that are higher for those who have higher gratitude to God.  This point I am getting at may sound picky, but it is not. Psychologically, SWB does not equal RWB, so the language the article should not sound as if it does by saying that higher gratitude to God predicts higher SWB (as if to imply total SWB).  SWB is the combination of RWB and EWB.  (By the way, EWB scores should be referred to as degree of Existential Well-Being, not as "secular well-being", because EWB is also part of some people's spirituality.  The author should not use language that suggests that only RWB is "spiritual", because it is incorrect according to a vast amount of prior research.)  In sum, the above noted misuse of the technicalities of SWB research and terminology needs to be corrected. I think making this correction is not optional.  I checked "minor revision" because it is small and easy to correct it. But it is major in terms of not saying misleading things, which can only cause confusion in the larger body of literature, to the readers.

Author Response

R1 was was primarily concerned about our use of the term "spiritual well-being." We used this term because our outcome variables were not just the Spiritual Well-being Scale, but also the Revised Nearness to God Scale and the Santa Clara Strength of Religious Commitment Scale. Gratitude to God predicted all three of these variables over time. We felt that "spiritual well-being" better captured all of these scales, as we defined in the opening paragraphs. However, we see R1's point that the term "spiritual well-being" might be somewhat misleading, and thus we have changed references throughout the manuscript to "religious well-being", and on occasion, "religious/spiritual well-being." 

We also agree with R1 that our term "secular well-being measures" may not communicate very effectively, because for many individuals these well-being measures probably include spiritual well-being. Thus we have changed "secular well-being" to "general well-being" throughout the manuscript.